# Trajectory Inference via Mean-field Langevin in Path Space

**Lénaïc Chizat**[*,†]    **Stephen Zhang**[*,‡]    **Matthieu Heitz**[§]    **Geoffrey Schiebinger**[§]

## Abstract

Trajectory inference aims at recovering the dynamics of a population from snapshots of its temporal marginals. To solve this task, a min-entropy estimator relative to the Wiener measure in path space was introduced in Lavenant et al. [1], and shown to consistently recover the dynamics of a large class of drift-diffusion processes from the solution of an infinite dimensional convex optimization problem. In this paper, we introduce a grid-free algorithm to compute this estimator. Our method consists in a family of point clouds (one per snapshot) coupled via Schrödinger bridges which evolve with noisy gradient descent. We study the mean-field limit of the dynamics and prove its global convergence to the desired estimator. Overall, this leads to an inference method with end-to-end theoretical guarantees that solves an interpretable model for trajectory inference. We also present how to adapt the method to deal with mass variations, a useful extension when dealing with single cell RNA-sequencing data where cells can branch and die.

## 1 Introduction

Trajectory inference aims at recovering the dynamics of a population of particles given samples from its temporal marginals at various time-points. This problem arises notably in the analysis of single-cell RNA-sequencing data [2–4], where one has access—via a destructive measurement process—to the cell state of samples from a large population of cells that evolve in time. In this application, one would like to recover the overall dynamics of the population as well as the trajectories of individual cells so as to improve our understanding of certain biological processes, such as embryonic development or tumor progression.

Trajectory inference can be cast as a regression problem where the unknown is the law of a continuous stochastic process. Let $\mathcal{X}$ be the ambient space containing the particles and $\Omega := \mathcal{C}([0,1], \mathcal{X})$ the path-space, i.e. the set of all possible continuous trajectories over the time interval $[0,1]$. We consider the problem of recovering a probability distribution over paths $P \in \mathcal{P}(\Omega)$, i.e. the law of a continuous stochastic process $(X_t)_{t\in[0,1]}$. For the inference to be well-behaved, one should look for a distribution $P$ such that its time marginals are consistent with the observed snapshots and such that the overall dynamics it represents satisfies some notion of regularity.

The problem of trajectory inference has received significant attention over the past several years. However, while pioneering work has focused on creating new methods, the theoretical treatment has remained limited. One class of methods focuses on recovering a potential energy landscape that best fits the observed marginals. For example, Hashimoto *et al.* [5] encode the potential with a neural network, TrajectoryNet [3] encodes the potential as a neural ODE, and JKOnet [6] encodes the potential with a neural network architecture based on the JKO scheme [7] and input convex neural

---

[*]Joint first authors.
[†]Institute of mathematics, EPFL, Switzerland
[‡]University of Melbourne, Victoria, Australia
[§]University of British Columbia, Canada

36th Conference on Neural Information Processing Systems (NeurIPS 2022).

networks. However, these current approaches are all nonconvex, and therefore it can be difficult to establish rigorous guarantees.

Some guarantees have been established for the equilibrium case by Weinreb *et al.* [8], and recently consistency was established for the non-equilibrium case by Lavenant *et al.* [1], who, as a regularity prior, penalize the entropy of $P$ relative to the Wiener measure on $\Omega$ (i.e. the law of the Brownian motion). Moreover, Bunne *et al.* have recently shown how to produce an improved reference process, beyond ordinary Brownian motion [9].

In this work we focus on the estimator introduced by Lavenant *et al.* [1], which reconstructs the process $P$ by minimizing the entropy relative to the Wiener measure over $\Omega$. By trading off data fitting with regularization, this approach generalizes the approach of Waddington-OT [2] to datasets with many time-points yet possibly few samples per time-point, which is required for consistency [1], yields better performance in practice [10], and is achievable through single-embryo profiling [11]. While the approach of Lavenant *et al.* leads to a consistent estimator, it also leads to a challenging infinite dimensional convex optimization problem. It is tackled in the original paper by discretizing the space and then applying convex optimization methods. The goal of the present paper is to show that the specific structure of this estimator makes it amenable to a newly introduced class of grid-free stochastic methods called *Mean-Field Langevin* (MFL) dynamics [12, 13], a non-linear generalization of Langevin dynamics which enjoys quantitative global convergence guarantees [14, 15]. Instantiated in our context, this method consists in a family of point clouds (one per snapshot) coupled via entropy regularized optimal transport, a.k.a. Schrödinger bridges [16], which evolve with noisy gradient descent. Intuitively, these dynamics can be interpreted as a (non-linear) Langevin diffusion over the path space $\Omega$. Our approach leads to an inference method with end-to-end theoretical guarantees that solves an interpretable model for trajectory inference. We illustrate on Figure 1 the recovered estimator and on Figure 2 the MFL optimization dynamics.

**Organization of the paper**    In Section 2, we present the estimator of Lavenant et al. [1]. The heart of our theoretical contributions is in Section 3 where we introduce the MFL dynamics and show its quantitative convergence towards the min-entropy estimator at an exponential rate. Numerical experiments are shown in Section 4.

**Notation and blanket assumptions**    For two probability measures $\mu, \nu$, their relative entropy is $H(\mu|\nu) = \int \log(\mathrm{d}\mu/\mathrm{d}\nu)\mathrm{d}\mu$ if $\mu \ll \nu$ and $+\infty$ otherwise. For $n \in \mathbb{N}$, let $[n] := \{1, \ldots, n\}$. Throughout, the ambient space $\mathcal{X}$ is a compact convex subset of $\mathbb{R}^d$ or the $d$-dimensional torus $\mathbb{T}^d$ and $(B_t)_t$ is a Brownian motion. The path space is $\mathcal{C}([0,1]; \mathcal{X})$ and laws on path space are noted by capital letters, $P$ or $R$. We denote by $P_{t_1, \ldots, t_T} = (e_{t_1}, \ldots, e_{t_T})_{\#} P$ the marginal of such laws under the (joint) evaluation maps $e_t : \omega \mapsto \omega(t)$. We use boldface letters for families of probability measures $\boldsymbol{\mu} \in \mathcal{P}(\mathcal{X})^T$.

## 2   Min-Entropy Estimator in Path Space

### 2.1   Trajectory Inference as Stochastic Process Inference

**Model of population dynamics**    The first step of any inference task is to determine a prior on the ground truth. For population dynamics with mass conservation, a natural prior is given by drift-diffusion processes. We thus model the population dynamics as

$$\mathrm{d}X_t = -\nabla \Psi(t, X_t)\mathrm{d}t + \sqrt{\tau}\mathrm{d}B_t, \qquad \mathrm{Law}(X_0) = \mu_0, \qquad t \in [0,1], \tag{1}$$

where $\tau > 0$ is the temperature/diffusivity, assumed known, and $\Psi \in \mathcal{C}^2([0,1] \times \mathcal{X})$ is unknown (in case $\mathcal{X}$ has boundaries, one should also introduce a reflection term, that we ignore in this section). As discussed in prior works [5, 1], it is hopeless to recover the divergence-free component of the drift: it is thus natural to assume that the drift is given by the gradient of a function $\Psi$, called the *Waddington potential* or *epigenetic landscape* in the context of cell development. The noise level $\tau > 0$ models the inherent randomness of the ground truth process. It turns out that this noise also has a favorable effect on the algorithm we develop here. This model can be extended to allow particles to branch and die [17], this extension is discussed in Section 4.2.

**Model of measurements**    Let $0 \le t_1 < \cdots < t_T \le 1$ be a family of measurement times. In contrast to the classical field of inference for stochastic processes [18] where each realization $(X_t)_{t \in [0,1]}$

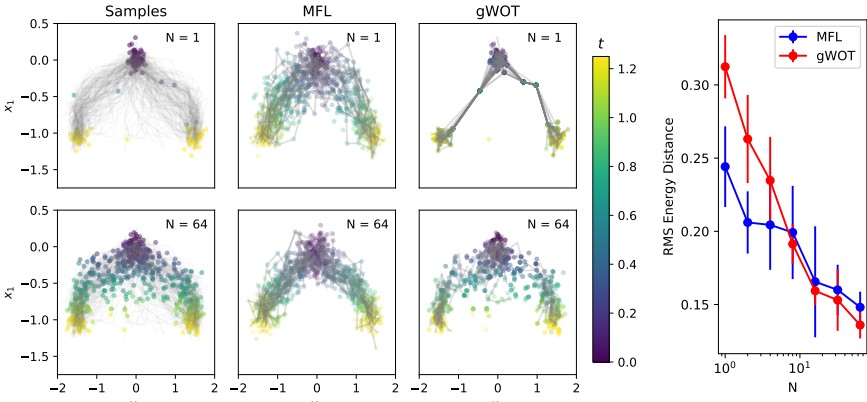

Figure 1: (left) Sampled time-series data and reconstructions by the proposed Mean-Field Langevin dynamics (MFL) and Global Waddington-OT [1] (gWOT). Observed and reconstructed marginals are colored by the measurement time $t_i$, and illustrative sample paths are overlaid in grey. We compare a scenario with limited data at intermediate time-points ($N = 1$) to a uniformly sampled scenario ($N = 64$). (right) RMS Energy Distance to the ground truth over marginals for $N = 2^0, \ldots, 2^6$, illustrating the robustness of MFL to the low-sample regime. See §4.1 for details.

is observed $T$ times (with $T$ large), in trajectory inference each realization is only observed once, i.e. we observe $(X_{t_i,j})_{i\in[T],j\in[n_i]}$ with independence for each couple $(i,j)$. The measurements are summarized by the family of snapshots $\hat{\mu}_{t_i} = \frac{1}{n_i}\sum_{j=1}^{n_i}\delta_{X_{t_i,j}} \in \mathcal{P}(\mathcal{X})$ for $i \in [T]$, where $n_i \geq 1$ is the number of particles observed at time $t_i$.

**Inference problem** The general goal is to recover the law $P = \text{Law}(X) \in \mathcal{P}(\Omega)$ of the stochastic process given the snapshots $(\hat{\mu}_{t_1}, \ldots, \hat{\mu}_{t_T})$. This object $P$ contains all the information about $X$: in particular it contains the marginals $P_t$ of the process and the transition probability between each family of marginals $P_{t_1,\ldots,t_T}$. Note that, as discussed in the introduction, some works [3, 5, 6] focus on directly recovering the Waddington potential $\Psi$ within a parameterized class of functions. This is a different problem, with encouraging empirical results but weaker known theoretical guarantees.

## 2.2 Min-Entropy Estimator

Let $W^\tau \in \mathcal{P}(\Omega)$ be the law of the (reflected) Brownian motion on $\mathcal{X}$ with temperature $\tau$ and uniform initialization. The min-entropy estimator[5] introduced in [1], is defined as the unique minimizer $R^*$ of the functional $\mathcal{F}: \mathcal{P}(\Omega) \to \mathbb{R}$ defined as

$$\mathcal{F}(R) := \frac{1}{\lambda}\sum_{i=1}^{T}\Delta t_i \text{Fit}_\sigma(R_{t_i}|\hat{\mu}_{t_i}) + \tau H(R|W^\tau) \tag{2}$$

where $\lambda > 0$ is the regularization strength, $\Delta t_i := (t_{i+1} - t_{i-1})/2$ (with the convention $t_{-1} = 0$ and $t_{T+1} = 1$) and $\text{Fit}_\sigma$ is a divergence functional that quantifies how much $R_{t_i}$ and $\hat{\mu}_{t_i}$ differ, parameterized by a bandwidth $\sigma > 0$. For a particular choice of fitting functional (see below), the authors prove the following result:

**Theorem 2.1** (Consistency, [1]). *If $(t_i)_{i\in[T]}$ becomes dense in $[0,1]$ as $T$ grows, then*

$$\lim_{\lambda,\sigma\to 0}\lim_{T\to\infty} R^* = P \qquad \text{weakly, almost surely.}$$

In this paper, we consider the following data fitting term

$$\text{Fit}_\sigma(R_t, \hat{\mu}_t) := \int -\log\left[\int \exp\left(-\frac{\|x-y\|^2}{2\sigma^2}\right)dR_t(x))\right]d\hat{\mu}_t(y) \tag{3}$$

$$= H(\hat{\mu}_t|R_t * g_\sigma) + H(\hat{\mu}_t) + C \tag{4}$$

---

[5]We propose this name because $R^*$ is the probability measure in path space with the smallest entropy relative to the Wiener measure among all those that fit at least equally well the observations.

where $g_\sigma(x) = (2\pi\sigma^2)^{-\frac{d}{2}} e^{-\|x\|^2/(2\sigma^2)}$ is the Gaussian density and $C > 0$ is a constant. Eq. (3) defines $\mathrm{Fit}_\sigma$ and is always well-defined and finite, but Eq. (4) only makes sense when $H(\hat{\mu}_t) < \infty$. This is the negative log-likelihood under the noisy observation model $\hat{X}_{t_i,j} = X_{t_i,j} + \sigma Z_{i,j}$ where $\hat{X}_{t_i,j}$ is the observation and $Z_{i,j} \overset{iid}{\sim} \mathcal{N}(0, I)$ the noise. Intuitively, when minimizing this *soft-min* objective over measures $R_t$ which are mixtures of Dirac masses, each of the observed particles from $\hat{\mu}_t$ tends to attract the particles from $R_t$ that are the closest—even if they are far away in an absolute sense—but barely influences those that are the farthest.

This data-fitting functional $\mathrm{Fit}_\sigma$ slightly differs from $H(g_\sigma * \hat{\mu}_t | R_t)$, which is the one for which Thm. 2.1 has been established in [1], but it preserves its key properties, namely joint convexity in $(R_t, \hat{\mu}_t)$ and linearity in $\hat{\mu}_t$. Our choice for $\mathrm{Fit}_\sigma$ is motivated by a more favorable behavior when $R_t$ is a discrete measure and by its natural statistical interpretation. In this paper, we focus on the optimization aspects and leave the statistical analysis of the estimator $R^*$ with this specific choice of $\mathrm{Fit}_\sigma$ for future work. Let us also mention that Thm. 2.1 has been established for $\mathcal{X}$ a compact manifold without boundary, while under our assumptions $\mathcal{X}$ may have boundaries.

We now turn to the presentation of the algorithm. The data model of Eq. (1) was introduced only to motivate the min-entropy estimator via Thm. 2.1, and plays no role in the rest of the paper.

# 3 Optimization Method: Mean Field Langevin Dynamics

In order to compute the min-entropy estimator $R^* \in \mathcal{P}(\Omega)$, one needs to minimize an entropy regularized functional over the space of paths $\mathcal{F} : \mathcal{P}(\Omega) \to \mathbb{R}$ which is of the form

$$\mathcal{F}(R) = \mathrm{Fit}(R_{t_1}, \ldots, R_{t_T}) + \tau H(R | W^\tau) \tag{5}$$

where we have posed $\mathrm{Fit}(R_{t_1}, \ldots, R_{t_T}) := \frac{1}{\lambda} \sum_{i=1}^T \Delta t_i \mathrm{Fit}_\sigma(R_{t_i} | \hat{\mu}_{t_i})$. This is the sum of a convex functional Fit and the relative entropy with respect to $W^\tau$. If, instead of $\mathcal{P}(\Omega)$, the optimization space was $\mathcal{P}(\mathcal{X})$, a problem with this structure could be solved by Mean-Field Langevin (MFL) dynamics. More explicitly, MFL dynamics are designed to minimize problems of the form $F(\mu) = G(\mu) + \tau H(\mu)$ where $G : \mathcal{P}(\mathcal{X}) \to \mathbb{R}$ is "smooth" and $H$ is minus the differential entropy. They enjoy global convergence guarantees when $G$ is convex. Considering for some $m \in \mathbb{N}^*$ the noisy gradient descent on the function $(x_1, \ldots, x_m) \mapsto G(\frac{1}{m} \sum_{i=1}^m \delta_{x_i})$, MFL dynamics are obtained in the mean-field $m \to \infty$ and vanishing step-size limit, see Section 3.2 for details.

Inspired by this parallel, we now design a drift-diffusion dynamics in path space that converges to the minimizer of $\mathcal{F}$. The main idea is a reformulation of the problem as a "reduced" problem over $\mathcal{P}(\mathcal{X})^T$ with a structure amenable to MFL dynamics.

## 3.1 Reduced Formulation

We first introduce the reduced functional $F$, and then state its connection to $\mathcal{F}$ in Thm. 3.1.

For $\mu, \nu \in \mathcal{P}(\mathcal{X})$, let $\Pi(\mu, \nu)$ be the set of transport plans between $\mu$ and $\nu$, that is, probability measures on $\mathcal{X} \times \mathcal{X}$ with respective marginals $\mu$ and $\nu$. The entropy regularized optimal transport cost between $\mu$ and $\nu$ is defined, for some $\tau_i > 0$, as

$$T_{\tau_i}(\mu, \nu) := \min_{\gamma \in \Pi(\mu,\nu)} \int c_{\tau_i}(x, y) \mathrm{d}\gamma(x, y) + \tau_i H(\gamma | \mu \otimes \nu) = \min_{\gamma \in \Pi(\mu,\nu)} \tau_i H(\gamma | p_{\tau_i} \mu \otimes \nu). \tag{6}$$

where $p_t(x, y)$ is the transition probability density of the (reflected) Brownian motion on $\mathcal{X}$ over the time interval $[0, t]$ – or equivalently the heat kernel with no-flux boundary conditions – and $c_{\tau_i}(x, y) := -\tau_i \log(p_{\tau_i}(x, y))$. In numerical experiments we use the approximation suggested by Varadhan's formula when $\tau_i$ is small: $c_{\tau_i}(x, y) \approx \frac{1}{2} \|y - x\|^2$ (up to an additive constant which is irrelevant when one is interested in minimizers only) [19]. This optimization problem is also known as the *Schrödinger bridge problem*: its solution gives the most likely evolution of a cloud of particles following a Brownian motion, conditioned on being distributed as $\mu$ at $t = 0$ and $\nu$ at $t = 1$ [16]. Some background on this problem is given in Appendix A, including the definition of the Schrödinger potentials $(\varphi, \psi)$ used hereafter. Consider the function $G : \mathcal{P}(\mathcal{X})^T \to \mathbb{R}$ defined for

$\boldsymbol{\mu} = (\boldsymbol{\mu}^{(1)}, \ldots, \boldsymbol{\mu}^{(T)})$ that represents a family of $T$ reconstructed temporal marginals, by

$$G(\boldsymbol{\mu}) := \text{Fit}(\boldsymbol{\mu}) + \sum_{i=1}^{T-1} \frac{1}{t_{i+1} - t_i} T_{\tau_i}(\boldsymbol{\mu}^{(i)}, \boldsymbol{\mu}^{(i+1)}) \tag{7}$$

where $\tau_i := (t_{i+1} - t_i)\tau$. We now introduce the *reduced objective* $F : \mathcal{P}(\mathcal{X})^T \to \mathbb{R}$, defined as

$$F(\boldsymbol{\mu}) := G(\boldsymbol{\mu}) + \tau H(\boldsymbol{\mu}) \tag{8}$$

where $H(\boldsymbol{\mu}) = \sum_{i=1}^{T} \int \log(\boldsymbol{\mu}^{(i)}(x)) \mathrm{d}\boldsymbol{\mu}^{(i)}(x)$ is minus the differential entropy of the family of measures $\boldsymbol{\mu}$.

The next result makes the link between minimizing $\mathcal{F}$, the objective in path space (5), and $F$, the reduced objective (8). It can be interpreted as a Representer Theorem for the min-entropy estimator.[6] This theorem is straightforward to deduce from [20, Cor. 3.5], but the specific form of $F$ that we use here is central for our subsequent algorithmic developments.[7]

**Theorem 3.1** (Representer Theorem). *Let* $\text{Fit} : \mathcal{P}(\mathcal{X})^T \to \mathbb{R}$ *be any function.*

   (i) *If $\mathcal{F}$ admits a minimizer $R^*$ then $(R^*_{t_1}, \ldots, R^*_{t_T})$ is a minimizer for $F$.*

   (ii) *Conversely, if $F$ admits a minimizer $\boldsymbol{\mu}^* \in \mathcal{P}(\mathcal{X})^T$ then a minimizer $R^*$ for $\mathcal{F}$ is built as*

$$R^*(\cdot) = \int_{\mathcal{X}^T} W^\tau(\cdot|x_1, \ldots, x_T) \mathrm{d}R_{t_1, \ldots, t_T}(x_1, \ldots, x_T)$$

   *where $W^\tau(\cdot|x_1, \ldots, x_T)$ is the law of $W^\tau$ conditioned on passing through $x_1, \ldots, x_T$ at times $t_1, \ldots, t_T$ respectively and $R_{t_1, \ldots, t_T}$ is the composition of the transport plans $\gamma_{i,i+1}$ which are optimal in the definition of $T_{\tau_i}(\boldsymbol{\mu}^{*(i)}, \boldsymbol{\mu}^{*(i+1)})$, for $i = 1, \ldots, T - 1$.*

The composition of the transport plans is obtained as

$$R_{t_1, \ldots, t_T}(\mathrm{d}x_1, \ldots, \mathrm{d}x_T) = \gamma_{1,2}(\mathrm{d}x_1, \mathrm{d}x_2)\gamma_{2,3}(\mathrm{d}x_3|x_2) \ldots \gamma_{T-1,T}(\mathrm{d}x_T|x_{T-1})$$

where we have introduced the conditional probability (a.k.a. "disintegrations") characterized by $\gamma_{i,i+1}(\mathrm{d}x_i, \mathrm{d}x_{i+1}) = \gamma_{i,i+1}(\mathrm{d}x_{i+1}|x_i)\mu_i(\mathrm{d}x_i)$. In probabilistic terms, the equality in $(ii)$ can be understood as saying that conditional on passing through $(x_1, \ldots, x_T)$ at times $(t_1, \ldots, t_T)$, the paths of $R^*$ are Brownian bridges with diffusivity $\tau$. The proof of Theorem 3.1 is given in Appendix B.

The importance of the reduced problem comes from the following facts:

• The optimization space has been "reduced" from $\mathcal{P}(\Omega)$ to $\mathcal{P}(\mathcal{X})^T$. This reduction is enabled by the Markovian property of $W^\tau$. Moreover, Theorem 3.1 gives an explicit method to construct a minimizer for $\mathcal{F}$ from a minimizer for $F$ and the associated optimal transport plans $(\gamma_{i,i+1})_{i \in [T-1]}$. This fact was already exploited in Lavenant et al. [1].

• The objective function $F$ is the sum of two terms: a "smooth" function $G$ and a differential entropy term $\tau H$. This is precisely the structure tackled by MFL dynamics. Observe how the entropy in path space $H(P|W^\tau)$ is split into two parts: the Schrödinger bridges $T_\tau$, included in the "smooth" term $G$, and minus the differential entropy $H$.

Let us now describe some useful properties of $G$ and $F$. Hereafter, the *first-variation* of $G : \mathcal{P}(\mathcal{X})^T \to \mathbb{R}$ at $\boldsymbol{\mu}$ is the unique (up to an additive constant) function $V[\boldsymbol{\mu}] \in \mathcal{C}(\mathcal{X})^T$ such that for all $\boldsymbol{\nu} \in \mathcal{P}(\mathcal{X})^T$,

$$\lim_{\epsilon \downarrow 0} \frac{1}{\epsilon}\big(G((1-\epsilon)\boldsymbol{\mu} + \epsilon\boldsymbol{\nu}) - G(\boldsymbol{\mu})\big) = \sum_{i=1}^{T} \int V^{(i)}[\boldsymbol{\mu}](x)\mathrm{d}(\boldsymbol{\nu} - \boldsymbol{\mu})^{(i)}(x). \tag{9}$$

**Proposition 3.2.** *The function $G$ is convex separately in each of its input (but not jointly), weakly continuous and its first-variation is given for $\boldsymbol{\mu} \in \mathcal{P}(\mathcal{X})^T$ and $i \in [T]$ by*

$$V^{(i)}[\boldsymbol{\mu}] = \frac{\delta\text{Fit}}{\delta\boldsymbol{\mu}^{(i)}}[\boldsymbol{\mu}] + \frac{\varphi_{i,i+1}}{t_{i+1} - t_i} + \frac{\psi_{i,i-1}}{t_i - t_{i-1}}, \quad \frac{\delta\text{Fit}}{\delta\boldsymbol{\mu}^{(i)}}[\boldsymbol{\mu}] : x \mapsto -\frac{\Delta t_i}{\lambda}\int \frac{g_\sigma(x-y)}{(g_\sigma * \boldsymbol{\mu}^{(i)})(y)}\mathrm{d}\hat{\mu}_{t_i}(y)$$

---

[6] We stretch a bit the term "representer theorem" which is usually reserved to finite-dimensional reductions.

[7] The key difference with [1, Prop. B.1] is that here $T_\tau$ involves the entropy of $\gamma$ relative to the product measure (instead of the volume measure), which makes $H(\boldsymbol{\mu})$ appear with a *positive* sign in $F$ (instead of negative).

*where $(\varphi_{i,j}, \psi_{i,j}) \in \mathcal{C}^\infty(\mathcal{X})$ are the Schrödinger potentials for $T_{\tau_i}(\boldsymbol{\mu}^{(i)}, \boldsymbol{\mu}^{(j)})$, with the convention that the corresponding term vanishes when it involves $\psi_{1,0}$ or $\varphi_{T,T+1}$. The function $F$ is jointly convex and admits a unique minimizer $\boldsymbol{\mu}^*$, which has an absolutely continuous density (again denoted by $\boldsymbol{\mu}^*$) characterized by*

$$(\boldsymbol{\mu}^*)^{(i)} \propto e^{-V^{(i)}[\boldsymbol{\mu}^*]/\tau}, \quad \text{for } i \in [T].$$

From now on, we thus focus on minimizing the reduced functional $F$ in Eq. (8), keeping in mind that this is sufficient to minimize $\mathcal{F}$, which is our main goal.

### 3.2 Mean-Field Langevin Dynamics

As previously mentioned, the MFL dynamics is natural when it comes to minimize functionals of the form $F_\epsilon = G + (\tau + \epsilon)H$. Here, we are increasing the entropy factor by $\epsilon > 0$ (recalled as an index of $F$) which will be useful to obtain convergence guarantees because $G$ is not convex but $F_0 = G + \tau H$ is. Using the first-variation $V[\boldsymbol{\mu}]$ of $G$ given in Prop. 3.2, the MFL dynamics is defined as the solution of the following non-linear SDE of McKean-Vlasov type, for $s \geq 0$:

$$\begin{cases} dX_s^{(i)} = -\nabla V^{(i)}[\boldsymbol{\mu}_s](X_s^{(i)})\mathrm{d}s + \sqrt{2(\tau + \epsilon)}\mathrm{d}B_s^{(i)} + \mathrm{d}\Phi_s^{(i)}, \quad \text{Law}(X_0^{(i)}) = \boldsymbol{\mu}_0^{(i)} \\ \boldsymbol{\mu}_s^{(i)} = \text{Law}(X_s^{(i)}), \quad i \in [T] \end{cases} \tag{10}$$

where $\mathrm{d}\Phi_s^{(i)}$ is the boundary reflection in the sense of Skorokhod problem, see [21, 22] (this term is not needed when $\mathcal{X}$ is the $d$-torus). Beware that the pseudo-time $s$ of the optimization dynamics in Eq. (10) should not be mistaken with the pseudo-time $t$ of the process in Eq. (1), which is now represented by the discrete exponents $i \in [T]$.

The family of laws $(\boldsymbol{\mu}_s)_{s \geq 0}$ of this stochastic process are characterized by the following system of PDEs (understood in the sense of distributions, with no-flux boundary conditions):

$$\partial_s \boldsymbol{\mu}_s^{(i)} = \nabla \cdot (\nabla V^{(i)}[\boldsymbol{\mu}_s]\boldsymbol{\mu}_s^{(i)}) + (\tau + \epsilon)\Delta \boldsymbol{\mu}_s^{(i)}, \quad i \in [T] \tag{11}$$

which are coupled via the quantity $\nabla V^{(i)}[\boldsymbol{\mu}_s]$. The link between (10) and (11) follows from Ito-Tanaka's formula, see e.g. [23, Lem. C3]. This is a multi-species PDE where each of the species $\boldsymbol{\mu}^{(i)}$ attempts to minimize $\frac{\Delta t_i}{\lambda}\text{Fit}_\sigma(\cdot|\hat{\mu}_{t_i}) + (\tau + \epsilon)H$ via a drift-diffusion dynamics, and is connected to $\boldsymbol{\mu}^{(i-1)}$ and $\boldsymbol{\mu}^{(i+1)}$ via Schrödinger bridges.

### 3.3 Quantitative Convergence

Let us now state the main convergence result proved in Appendix C.

**Theorem 3.3** (Convergence). *Let $\boldsymbol{\mu}_0 \in \mathcal{P}(\mathcal{X})^T$ be such that $F(\boldsymbol{\mu}_0) < \infty$. Then for $\epsilon \geq 0$, there exists a unique solution $(\boldsymbol{\mu}_s)_{s \geq 0}$ to the MFL dynamics (11). For $\epsilon > 0$, $\mathcal{X}$ the $d$-torus and moreover assuming that $\mu_0$ has a bounded absolute log-density, it holds*

$$F_\epsilon(\boldsymbol{\mu}_s) - \min F_\epsilon \leq e^{-Cs}\big(F_\epsilon(\boldsymbol{\mu}_0) - \min F_\epsilon\big).$$

*where $C = \beta e^{-\alpha/\epsilon}$ for some $\alpha, \beta > 0$ independent of $\boldsymbol{\mu}_0$ and $\epsilon$. Moreover, taking a smooth time-dependent $\epsilon_s$ that decays asymptotically as $\tilde{\alpha}/\log(s)$ for some $\tilde{\alpha} > \alpha$, it holds $F_0(\boldsymbol{\mu}_s) - F_0(\boldsymbol{\mu}^*) \lesssim \log(\log(s))/\log(s) \to 0$ and $\boldsymbol{\mu}_s$ converges weakly to the min-entropy estimator $\boldsymbol{\mu}^*$.*

We can make the following comments:

- Under these assumptions, and with $\epsilon$ fixed, the convergence of $\boldsymbol{\mu}_s$ to the minimizer of $F_\epsilon$ in relative entropy and in Wasserstein distance, also holds, with the same rate [14, 15]. Note that our convergence argument slightly differ from these works, in that the functional $G$ is not convex itself, and the noise/diffusion term is used both to convexify the objective (since $G + \tau H$ is convex) and to make the dynamics converge (via the additional $\epsilon H$ term).

- By Thm. 3.1, one can map $(\boldsymbol{\mu}_s)_{s \geq 0}$ to a dynamics in $\mathcal{P}(\Omega)$: it is this dynamics which we refer to as "Mean-Field Langevin in Path Space" in the title.

- This result gives a convergence rate for an optimization dynamics in a non-convex landscape, so it is not even obvious to have global convergence. The convergence rate depends exponentially on $-1/\epsilon$, a standard drawback of Langevin-like dynamics in absence of log-concavity.

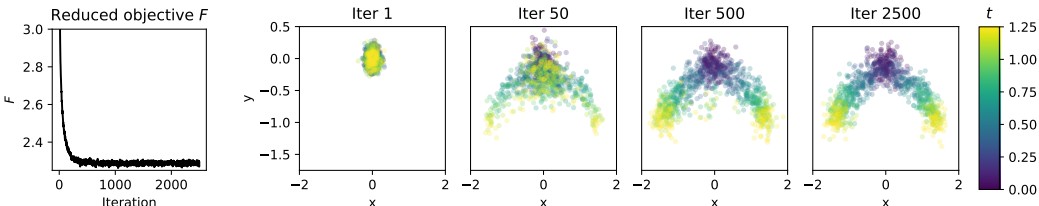

Figure 2: Optimization dynamics in the setting of Figure 1 with $N = 64$. (left) Evolution of the objective function $F(\boldsymbol{\mu}_s)$ (Eq. (8)) (right) Evolution of the reconstructed marginals $\hat{\boldsymbol{\mu}}^{(i)}[k]$ (Eq. (13)) with the iteration number $k$, starting from isotropic Gaussians at $k = 1$ and colored by measurement time $t_i$.

- The only point in the proof where the assumption that $\mathcal{X}$ is the $d$-torus is needed is the technical Lem. C.3 and we believe that the same convergence statement holds for $\mathcal{X}$ a convex compact domain of $\mathbb{R}^d$. Further extending Thm. 3.3 to the non-compact case would however raise more profound theoretical challenges (even the well-posedness of the MFL dynamics is not clear in this case).

**Summary of theoretical guarantees:** Overall, the MFL dynamics of Eq. (11) with simulated annealing ($\epsilon_s \to 0$) converges to the unique minimizer $\boldsymbol{\mu}^*$ of the reduced problem in Eq. (8), from which we can build by Thm 3.1 the min-entropy estimator $R^* \in \mathcal{P}(\Omega)$ which is optimal for (5). In this sense, our method leads to an inference method with end-to-end theoretical guarantees for trajectory inference. However, this statement is about an idealized dynamics which in practice has to be discretized in time and with particles. In the next section (§3.4), we will see that the resulting discretization error can be controlled, leveraging the long history of work on mean-field dynamics.

### 3.4 Discretization

In practice, an approximation of the MFL dynamics is obtained by running noisy gradient descent on the function $G_m : (\mathcal{X}^m)^T \to \mathbb{R}$ defined as

$$G_m(\hat{X}) := G(\hat{\boldsymbol{\mu}}_{\hat{X}}) \qquad \text{where} \qquad \hat{\boldsymbol{\mu}}_{\hat{X}}^{(i)} = \frac{1}{m} \sum_{j=1}^{m} \delta_{\hat{X}_j^{(i)}} \qquad (12)$$

where $m \in \mathbb{N}$ is be number of particles used to discretized each of the time marginals $\boldsymbol{\mu}^{(i)}$. Since it can be shown that $m \nabla_{X_j^{(i)}} G_m(\hat{X}) = \nabla V^{(i)}[\hat{\boldsymbol{\mu}}_{\hat{X}}](\hat{X}_j^{(i)})$ (see e.g. [15, Prop. 2.4]), this leads to the following update equations, for $k \geq 0$, $i \in [T]$ and $j \in [m]$:

$$\begin{cases} \hat{X}_j^{(i)}[k+1] = \hat{X}_j^{(i)}[k] - \eta \nabla V^{(i)}[\hat{\boldsymbol{\mu}}[k]](\hat{X}_j^{(i)}[k]) + \sqrt{2\eta(\tau + \epsilon)} Z_{j,k}^{(i)}, \quad \hat{X}_j^{(i)}[0] \overset{iid}{\sim} \boldsymbol{\mu}_0^{(i)} \\ \hat{\boldsymbol{\mu}}^{(i)}[k] = \frac{1}{m} \sum_{j=1}^{m} \delta_{\hat{X}_j^{(i)}[k]}, \quad i \in [T] \end{cases} \tag{13}$$

where $\eta > 0$ is a step-size, $Z_{j,k}^{(i)}$ are independent standard Gaussian variables and one should moreover project all particles on $\mathcal{X}$ at each step in case $\mathcal{X}$ has boundaries. The convergence of such a numerical scheme towards the MFL dynamics is standard in the mean-field literature [24, 23], and holds here thanks to the stability of $\nabla V$ which follows from [25].

In the related context of wide neural networks, quantitative bounds on the discretization error were studied in [26, Thm. 2]. It is shown that for $s = k\eta$, with high probability, $|G(\hat{\boldsymbol{\mu}}_k) - G(\boldsymbol{\mu}_s)| = \tilde{O}(e^{Cs}(m^{-1/2} + \eta^{1/2}))$ for some $C > 0$ independent of $k, \eta$ and $m$. Inspecting their proof, it can be seen that this rate depends on the rate at which $G(\mu)$ and $\nabla V[\mu]$ can be estimated from independent samples of $\mu$, which is studied in our case in [27, 28]. The exponential dependence in the pseudo-time $s$ of these bounds make them however not precise enough to obtain interesting long-time guarantees and we leave for future work a direct convergence analysis of the fully discretized MFL dynamics (note that [14, Cor. 2] gives a guarantee for the *time*-discretized MFL dynamics).

Each iteration of Eq. (13) requires to solve an entropic optimal transport problem in order to obtain $\nabla V$. As discussed in Appendix E, this can be solved to precision $\tilde{\epsilon}$ in time $O(m^2/(\tau_i \tilde{\epsilon}))$ using Sinkhorn's algorithm.

# 4 Numerical Experiments

## 4.1 Simulated data

In our first experiment, we compare the behaviour of the MFL dynamics to that of Global Waddington-OT (gWOT) [1] in a setting with few samples per time-point. Although both methods minimize a functional of the form (2), gWOT works on a fixed, discretized support: the union of all observed sample points. Thus, gWOT should perform poorly when the support set has missing regions or "gaps". To simulate this, we simulated a dataset with $N_i$ samples at time $t_i$, with $N_1 = N_{10} = 64$ and $N_i = N, 2 \leq i \leq 9$ for $N \in \{2^0, \ldots, 2^6\}$. The samples are drawn from the marginals of a bifurcating stochastic differential equation (see Appendix G for details).

In Figure 1(left) we illustrate two extreme cases: $N = 1$ (very few samples at intermediate timepoints) and $N = 64$ (uniform sampling over time) respectively, with $\lambda = 0.05, \lambda_{\text{gWOT}} = 0.0025$. Visually, it seems that the output of MFL is robust to the few-sample regime, with relatively little qualitative difference between the reconstructed trajectories for $N = 1, 64$. On the other hand, the performance of gWOT degrades visibly once the set of observed points is a poor reflection of the support of the underlying process.

To examine this quantitatively, we applied both MFL dynamics and gWOT for various values of $N$ and computed the root-mean-square (RMS) Energy Distance [29] over time to an approximate ground truth (see Appendix G). As shown in Figure 1(b), for large values of $N$ we observe that both MFL and gWOT perform similarly, but with MFL outperforming gWOT for small $N$ as expected.

Finally, Figure 2 illustrates the evolution of the reduced objective $F$ (8) and the reconstructed marginals $\boldsymbol{\mu}^{(i)}[k]$ over the course of MFL dynamics for $N = 64, \lambda = 0.05$. To estimate the entropy term $H(\boldsymbol{\mu})$, we used the Kozachenko-Leonenko nearest neighbour estimator [30, Section 6] (note that estimating the entropy is *not* needed in the algorithm; it is only used here to illustrate the convergence).

## 4.2 Dealing with Branching

In the context of single cell RNA-sequencing data it is crucial to be able to deal with the birth and death of cells, a.k.a. branching, which does not occur homogeneously in the domain $\mathcal{X}$. Assume that we dispose of a function $g \in \mathcal{C}^1([0,1] \times \mathcal{X})$ such that $g(t, x)$ is the prior knowledge about the instantaneous growth rate of the distribution of particles in $x$ at time $t$ in the model of Eq. (1). That is, in time $dt$ the probability of a particle $X_t$ dividing is $g(t, X_t)dt$ [17]. We would like to incorporate this knowledge, and also allow for additional mass variations to account for the inaccuracy of our prior $g$.

To this end, we proceed heuristically by simply replacing $T_{\tau_i}(\mu_{t_i}, \mu_{t_{i+1}})$ in Eq. (6) with the following "unbalanced" Schrödinger bridge problem

$$\min_{\gamma \in \mathcal{M}_+(\mathcal{X}^2)} \int c_{\tau_i}(x, y) d\gamma(x, y) + \rho H(\gamma_1 | \tilde{\mu}_{t_i}) + \rho H(\gamma_2 | \tilde{\mu}_{t_{i+1}}) + \tau_i H(\gamma | \tilde{\mu}_{t_i} \otimes \tilde{\mu}_{t_{i+1}}) \qquad (14)$$

where $\tilde{\mu}_{t_i} \propto \exp(-g(x)(t_{i+1} - t_i)/2)\mu_{t_i}$ and $\tilde{\mu}_{t_{i+1}} \propto \exp(g(x)(t_{i+1} - t_i)/2)\mu_{t_{i+1}}$ are probability measures, $\rho > 0$ is a parameter, $\mathcal{M}_+(\mathcal{X}^2)$ is the set of nonnegative measures on $\mathcal{X}$ and $\gamma_1, \gamma_2$ are the marginals of $\gamma$. This problem can be solved with a variant of Sinkhorn's algorithm [31, 32].

The rationale behind this formula is as follows: (i) the modifications $\tilde{\mu}_{t_i}$ of $\mu_{t_i}$ are intended to approximate the growth process $\partial_t \hat{\mu}_t = g(t, \cdot)\hat{\mu}_t$ over the time interval $[t_i, t_{i+1}]$ and (ii) we relax the marginal constraints, with a parameter $\rho > 0$, to account for the potential inaccuracy of our prior $g$. We illustrate on Figure 3 the practical advantage of taking into account branching in the algorithm.

**Simulated data experiment** Here we consider a modified version of the bistable process from [1, Figure 18] (see Appendix G for details). Figure 3 shows paths sampled from the ground truth, sample points, and the inferred trajectories. Since the lower potential well is closer to the initial condition,

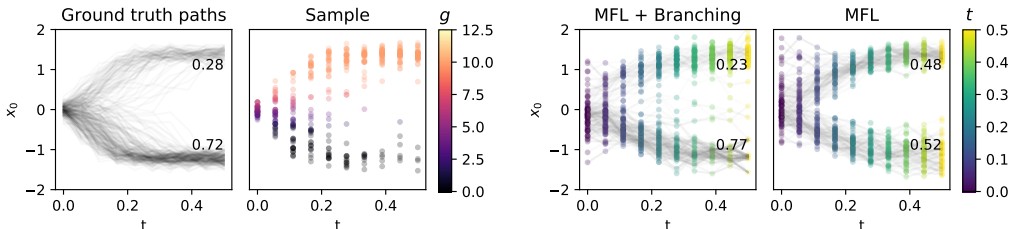

Figure 3: Accounting for branching rates allows for the separation of spatial dynamics from growth. (left) Ground truth paths and sample points. (right) Reconstruction produced by MFL dynamics with and without accounting for branching. Fraction of paths terminating in the upper (resp. lower) branch is annotated in the plot.

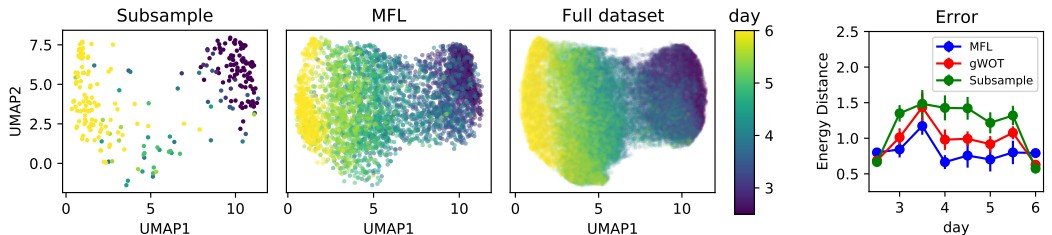

Figure 4: (left) Subsampled timepoints (Subsample) from the iPSC reprogramming dataset of [2], with 100 cells at the first and last timepoints and 10 cells at each intermediate timepoint, shown beside inferred marginals obtained using MFL dynamics and the full dataset for those timepoints. (right) Best performance as measured by Energy Distance of inferred marginals to the full dataset for MFL dynamics (Langevin) and Global Waddington-OT [1] (gWOT), with the subsampled timepoints (Sample) for reference.

we observe in the ground truth that $\approx 72\%$ of paths terminate in the lower branch. This effect is opposed by the high rate of branching in the upper branch, which ultimately results in more particles being sampled in the upper branch. By accounting for branching in the MFL dynamics we are able to reconstruct dynamics that isolate the effect of the potential, as evidenced by the comparable proportion (77%) of inferred paths ending near the lower branch. On the other hand, neglecting the presence of branching results in growth effects being confounded with the potential, and the proportion of paths ending near each branch are roughly equal.

### 4.3 Reprogramming dataset

We now consider the stem cell reprogramming dataset of [2], in which a population of differentiating cells were profiled over a time course using single-cell RNA sequencing. For the purpose of this example, we restrict our attention to days 2.5-6 inclusive making a total of 8 timepoints. As previously, we consider a regime where few samples are observed as snapshots of the time-series and we apply MFL dynamics to infer reconstructed marginals. We reason that if the underlying process is well described by a Schrödinger bridge (as was empirically validated in [2]), then MFL dynamics should be able to "improve" on the sample marginals. As a proxy for ground truth, we used the full dataset (consisting of 59154 cells over the 8 timepoints). From this, we subsampled timepoints consisting of 100 cells at days 2.5 and 6, and 10 cells at the remaining intermediate timepoints. After carrying out a series of preprocessing steps (see Appendix H for details), we applied MFL dynamics to the subsampled data to produce reconstructed marginals. Figure 4(left) shows an example of the sampled points, MFL reconstruction, and the full dataset.

A reconstruction "error" was then computed as the Energy Distance (20) of each reconstructed marginal to the corresponding snapshot in the full dataset. Figure 4(right) shows the error for each timepoint for each of MFL and gWOT, for the best performing parameters found in a parameter

sweep. We observe that both MFL and gWOT improve upon the subsampled snapshots, but MFL by a larger margin.

## 5   Conclusion

We introduced a grid-free numerical method for trajectory inference that computes the min-entropy estimator introduced in [1], with global and quantitative convergence guarantees in the mean-field limit. This method arises naturally when decomposing the optimization problem in a suitable way and, in practice, outperforms the fixed-grid method of [1].

Concerning limitations, our method shares those of the min-entropy estimator : since it does not incorporate fine structural prior on the structure of the Waddington potential (i.e. it is fully nonparametric), it may suffer from a limited statistical efficiency in high dimension. An interesting research direction is to quantify this statistical performance and to adapt our algorithm to learn the Waddington potential $\Psi$ in a structured parameterized class of functions, jointly with the law on paths $P$.

On a more abstract level, our main insight is that min-entropy problems in Wiener space can be solved via a multi-species diffusion dynamics coupled via Schrödinger bridges. This point of view raises interesting theoretical questions, e.g. can we rigorously interpret our dynamics as a diffusion in path space?

## Acknowledgements

GS was supported by a Career Award at the Scientific Interface from the Burroughs Wellcome Fund, a NFRF Exploration Grant, a NSERC Discovery Grant, and a CIHR Project Grant.

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
