# OpenReview forum: "Trajectory Inference via Mean-field Langevin in Path Space"
_NeurIPS.cc/2022/Conference — NeurIPS 2022 Accept_

### Official Review · Reviewer_qDLz · 2022-07-08

**Rating:** 8
**Confidence:** 5
**Soundness:** 4 excellent
**Presentation:** 4 excellent
**Contribution:** 4 excellent

**Summary:**

In this work authors study the problem of fitting a law on the path space induced by gradient SDE with additive and reflective Brownian motion to the set of prescribed time marginals. This is similar to multi-marginal OT with a difference the transport plan is restricted the be the law induced by an SDE.  The problem has been recently studied in Lavenant et al where, essentially, an estimator used in the current paper has been developed and analysed. Authors observed (which is not new) that one can translate entropic optimisation problem on the Wiener space (as in Lavenant et al.) to the optimisation problem on the product space of probability measures supported on the compact set \mathcal X (the compactness being a technical condition). This reformulation then open the avenue for mean-filed langevin dynamics (MFLD) (or gradient flow on W_2 space)  to be used as an optimisation ‘algorithm’ for which convergence along with the convergence speed can be deduced from recent works in the literature. Indeed mean-filed langevin dynamics has been recently studied by Mei et al (2018) and Hu et al (2019) where convergence has been proved rigorously. These result have been recently extended by Chizat 2022 and Nitanda et al 2022 that showed that under appropriate conditions convergence of MFLD is exponential. In the current paper authors showed, by assuming compactness of the space \mathcal X, that results of Chizat and Nitanda et al apply.

It’s worth remarking that conceptually this work is related to ‘Mean-Field Neural ODEs via Relaxed Optimal Control’ by Jabir at al 2019 and
‘Gradient Flows for Regularized Stochastic Control Problems’ by Siska et all 2020, where detailed analysis of coupled MFLD has been studied. In these works the cost function was slightly simpler and significant amount of regularisation was required to establish convergence.


**Questions:**

- It wasn’t clear where in the proof you used the fact that only SDE with drift of grade it type are needed? I would appreciate more discussion on that in the appendix
- When stating theorems please make sure assumptions are clearly stated
- In eqn 10 reflected BM is used. How do I know X lives in \mathcal X ? Don’t I need to consider Skorohod problem?
- Similarly to derive 11 from 10 Ito formula is used. Please Don’t you need Ito-Tanaka formula here?
- Proof of Lemma B.2 : for completes please provide more details rather then just citing [21] and [23]
- In deriving LSI compactness of \mathcal X is critical. Can you comment more about the constants how the diameter of D affects convergence ?
- please provide more details in the proof of Proposition D.1 rather than just citing other’s people work. That will make the paper easier to read.


**Ethics Review Area:**

["I don’t know"]

**Limitations:**

Mathematically main limitation is the compactness of \mathcal X, but the result is still very interesting.

**Strengths And Weaknesses:**

Strengths:
- Very interesting contribution tackling difficult problem of proving convergence rates for gradient flow algorithms for the inference on the path space.
-  The paper is (mainly) well written making highly technical material accessible to broad audience
- cell-RNA sequencing is a very promising area of applications for the ML models and I anticipate significant impact of this work on it

Weaknesses:
- The compactness of \mathcal X is not a natural assumption but seem necessary for the analysis to go through
- Some parts of the proofs are written in haste and I would appreciate more details (see comments bellow)
- When citing other works, especially in the proofs, please give exact detail of the results used rather than just the name of the paper…

---

> ### Author Response · Authors · 2022-08-02
> **We'll improve our treatment of the reflecting boundary conditions for the SDE and the redaction of the proofs**
>
> Thank you for the detailed comments! Concerning your summary, let us stress that it was not obvious from [Lavenant et al.’20] that the optimization problem of Eq.(5) could be tackled with MFL. We needed to rework their formulation to have entropy terms appearing with the correct sign. We consider this reformulation -- which leads to an optimization method that was not obvious at first sight -- as our main conceptual contribution. Concerning the technical details, we’ll provide more details and clarification for the proofs.
>
> **Questions**
>
> (1) To quote from the paper, “The data model of Eq. (1) was introduced only to motivate the min-entropy estimator via Thm. 2.1 [quoted from [Lavenant et al.]], and plays no role in the rest of the paper.” So this assumption does not intervene in our theory or proofs: our only assumption about the data is that we observe discrete empirical measures $\hat \mu_{t_i}$ supported on $\mathcal{X}$.
>
> (2) We’ll double check that all assumptions are clearly and correctly stated (did you have a specific theorem in mind with missing assumptions? Maybe this is about Thm. 2.1 which is quoted from another paper).
>
> (3) Yes we consider reflecting boundary conditions for SDE (10) (a.k.a. “Skorokhod problem”). We have now introduced explicitly the term of bounded variation that enforces the boundary condition in (10) for the sake of rigor.
>
> (4) Indeed, we need a formula that applies to reflected processes. Thank you for the correction. We have added an example of a precise reference where this derivation is proved using Ito-Tanaka’s formula.
>
> (5) We have added details about the part taken from [21]. As for the part coming from [23], it is a rather long proof by recursion and we use it exactly in the context of [23], so we prefer to cite the result.
>
> (6) In our original submission, we were providing a lower bound on $C$ (which gives the convergence rate of the form $e^{-Ct}$ in Thm.3.3) in terms of the parameters (including $D$). Unfortunately, with the corrected version of Thm. 3.3 it is a bit harder to track the constants and so we have decided to drop it. In any case, with our current approach, the lower bound cannot be better than $e^{-cst*D/\epsilon}$. One direction for future work is to derive the convergence rate in the non-compact case which requires, we believe, a very different viewpoint and would better reflect the behavior of the algorithm.
>
> (7) We have expanded some of the proofs, and we will continue working on making Prop.D.1 and the rest of the proofs more clear and self-contained.

---

### Official Review · Reviewer_mVx5 · 2022-07-11

**Rating:** 7
**Confidence:** 4
**Soundness:** 3 good
**Presentation:** 3 good
**Contribution:** 3 good

**Summary:**

This paper tackles the challenge of recovering the law of a stochastic process from sample observations of its marginals. The authors propose to approximate an existing, continuou-time, consistent estimator minimising a given regularised functional by another reduced, discrete-time estimator defined as the sum of a functional G and an entropy term H. The functional G is built from solutions to Schrodinger bridge problems between consecutive marginals. To obtain the reduced estimator, the authors consider a Langevin SDE of Mc-Kean-Vlasov type and show that its dynamics converge to the unique minimiser of the reduced optimisation problem. Numerical experiments are performed on simulated data from a bifurcating SDE as well as on real-world RNA-sequencing data.

**Questions:**

Is it possible to provide a more concrete interpretation of the integral in L. 156?

Can you provide additional details or comment on the time- and space-complexities of the proposed algorithm? In particular, how does it scale in the length and dimensionality of the paths?

How are the regularising constants chosen in the experiments?

In Figure 1, for large values of N, gWOT seems to start outperforming MFL. Does this trend continue if N is chosen in the order of 10^3, 10^4...?

Would the proposed methodology apply if one considered a different reference measure than Brownian motion (BM)? In particular, what would happen if one considered instead a non-Markovian reference process, for example fractional BM, or other processes with memory?

Regarding benchmarking as well as assessment of the success for the proposed method, have you considered divergences for probability measures supported on pathspace, such as the families of MMD distances studied in [1]?

[1] Salvi, C., Lemercier, M., Liu, C., Horvath, B., Damoulas, T., & Lyons, T. (2021). Higher order kernel mean embeddings to capture filtrations of stochastic processes. Advances in Neural Information Processing Systems, 34, 16635-16647.

**Limitations:**

As the authors highlight in their conclusion, the proposed algorithm suffers from the typical drawbacks of min-entropy estimators which makes it not scalable to high-dimensional problems.

**Strengths And Weaknesses:**

Recovering the law of a stochastic process from sporadic observations of its marginals is an important and difficult task, both at a theoretical and practical level, and this work makes good contributions in this direction. I enjoyed reading the paper and found its material original, well-written and the mathematical claims thoroughly justified in the appendix.

One possible weakness is the scalability of the proposed algorithm to high-dimensional and long trajectories, as noted by the authors themselves in their conclusion.

---

> ### Author Response · Authors · 2022-08-02
> **Hyper-parameter selection (and other points)**
>
> Thank you for your detailed comments!
>
> **Questions**
>
> (1) The integral in l.156 is, in a measure-theoretic sense, the disintegration of $R^*$ by the evaluation map $e$ at times $(t_1,...,t_T)$ (i.e. $e(\omega)=(\omega(t_1),\dots,\omega(t_T))$). Probabilistically, it can be understood as saying that conditional on passing through $(x_1, …, x_T)$ at times $(t_1,...,t_T)$, the paths of $R^*$ are Brownian bridges with diffusivity $\tau$. We have added this explanation.
>
> (2) [See our answer to Q3 of reviewer 2ZNK]
>
> (3) In the simulated experiments, the level of entropic regularization is determined by the ground truth diffusivity, i.e. $\tau * (t_{i+1} - t_i)$ between a pair of time-points $t_i$ and $t_{i+1}$. For the reprogramming dataset, the regularization level was chosen to be effectively $0.1 * E [(X_{t+1} - X_t)^2/2] $ between time-points $t_i$ and $t_{i+1}$, as is described in Section H of the supplement. This works out to be almost equivalent to the default value of $0.05*\text{median}[(X_{t+1} - X_t)^2]$ used in [Schiebinger et al., 2019]. In general for real data, the level of noise is not known and the problem of choosing the level of entropy regularization $\tau$ (as well as the data-fitting parameter $\sigma^2$) is related to that of bandwidth selection for kernel methods. For this, heuristics such as the mean and median criterion exist, see e.g. [Garreau et al. 2017].
>
>  [Garreau et al. 2017]: Garreau, D., Jitkrittum, W. and Kanagawa, M., 2017. Large sample analysis of the median heuristic. arXiv preprint arXiv:1707.07269.
>
> (4) This is a good question: we believe that this trend indeed continues when $N$ gets larger because our method estimates the marginal $\mu_t$ more diffused than they actually are. This is mainly owing to the data-fitting term which has a finite bandwidth parameter. Theory suggests that we should decrease the hyperparameter ($\lambda$ and $\sigma^2$) as $N$ increases, but in Fig. 1 only $\lambda$ is varied and $\sigma^2$ is kept fixed for simplicity (see supplement for details). In contrast, gWOT uses the same support as the input samples, which implicitly gives more strength to the data rather than the prior. If both $\lambda$ and $\sigma^2$ were allowed to vary with increasing $N$, we have reason to believe that this trend would not be observed.
>
> (5) Our method would work for any Markovian process as a reference (as long as its reversible measure has an explicit or tractable log-density). However, if the reference process is non-Markovian, then the “representer theorem” would not hold anymore and our approach would not apply; different ideas would be needed.
>
> (6) Thank you for the reference, we were not aware of such tools, which could be very useful indeed for this line of work! Note that in our case, the reconstructed stochastic process $R^*$ is characterized by the family of $T-1$ transport plans, which is a simpler object than a general stochastic process (SP). One point of difference between the provided reference and the present work is that [Salvi et al., 2021] consider a scenario where one has access directly to sample trajectories, whereas in our setting only population snapshots at fixed time-points are available; but their method could indeed be considered in synthetic experiments where the ground truth SP is known.

---

> > ### Comment · Reviewer_mVx5 · 2022-08-08
> > **Reply to Authors**
> >
> > Thank you for answering my questions. It would have been interesting to check your hypotheses about Q 4) with extra experiments, but this is probably an unreasonable request given the limited time of the rebuttal period. I still believe this paper provides a solid theoretical contribution to tackle the considered problem therefore I am keeping my score unchanged.

---

### Official Review · Reviewer_2ZNK · 2022-07-11

**Rating:** 5
**Confidence:** 3
**Soundness:** 3 good
**Presentation:** 2 fair
**Contribution:** 2 fair

**Summary:**

The authors consider the problem of finding the dynamics of a population from snapshots of its temporal marginals. They introducing an estimator which which is the sum of the goodness of fit to data term and entropy-based term (a.k.a. prior/regularizer). Instead of minimization of the entropy-regularized functional over the space of all paths, they introduce the “reduced” functional for optimization that includes additional entropy-regularized Optimal Transport problems between intermediate positions. The reformulated problem turns out to be equivalent to the initial problem. This follows from Theorem 3.1 which the authors prove (they call it the Representer theorem as its idea is analogous the idea of the original representer theorem). The resulting “reduced” problem is practically more tractable as it considers only a finite amount of time moments. The authors use Mean-Field Langevin (MFL) dynamics to approach the functional and recover the trajectory of the stochastic process, finding the optimal corresponding marginal distributions simultaneously. In the experimental session, they compare proposed approach with Global Waddington-OT (gWOT) in the simulated and real data experiments in small dimensions.

**Questions:**

(1) Consistency Theorem 2.1 seems to be formulated in [1] for a different than (3) functional (lines 110-111). Does the consistency hold for (3) which the authors use?

(2) How do you solve numerically PDE (11)?

(3) What is the computational complexity of the proposed method?

(4) How well does the method scale to higher dimensions? It would be great to see the error dynamics on some simulated dataset with the increase of the dimension.

(5) The unbalanced OT appears rapidly and out of nowhere. Is it true that all the theoretical results of the previous sections apply to formulation (14) instead of the original entropic balanced OT?


**Limitations:**

The authors sufficiently fully describe the limitations.

**Strengths And Weaknesses:**

**Strengths**

(1) While most other methods are based on recovering the Waddington potential directly, the current paper uses point of clouds (particles) to recover the desired dynamics.

(2) The theoretical part of the paper is interesting, specifically I would like to highlight the Theorem 3.1 in the spirit of Representer which reduces optimization over paths to optimization over finite products of measure spaces at observation times. Additionally, the paper proves the result on the exponential convergence of the functional which they optimize (Theorem 3.3).

**Weaknesses**

(1) The method is point-cloud-based (particle-based) and it is presumably not very scalable for high dimensions as it would require exponential amount of particles.

(2) The authors claim that existing approaches can be difficult to establish rigorous guarantees and establish guarantees for their method. However, when it comes to practice, it seems like in their method a large gap anyway appears between the theory and practice. For example, Thm. 3.3 is proved for measures, but in practice they are finitly approximated by particles (eq. (12)); entropy term is estimated by the nearest neighbor estimator (line 259); sinkhorn terms are also computed between discrete distributions – all these approximations introduce minor biases here-and-there which raises questions to which extent the proposed theory (Theorem 2.1, 3.3.) is still applicable and advantageous w.r.t. The other methods.

(3) The comparisons are a little bit limited – only the gWOT method is included. What about (some) of the other related methods? Specifically, seems like [6] or [9] might be relevant.

(4) The method has some hyperparameters which are not very clear how to choose.

(5) There is not clear explanation of the final algorithm with description of all the inputs, outputs, parameters, etc. Seems like these details can be collected from the main text and appendix, but this is a negative factor for reproducibility and readability.

---

> ### Author Response · Authors · 2022-08-02
> **Computational complexity bounds left to future work, but a priori no curse of dimensionality**
>
> Thank you for your detailed comments.
>
> **Weaknesses**
>
> (1) Although we do not study the challenging problem of particle discretization here, related results in the literature on mean-field Langevin dynamics suggest that that the method *does not* require an exponential number of particles. Instead we expect the error to decrease at a polynomial rate in the number of particles (independent of $d$) thanks to the stability of entropy regularized optimal transport (EOT) as discussed in Sec. 3.4.
>
> (2) Here, by guarantees, we mean consistency (i.e. non-quantitative) guarantees, e.g. as the number of particles goes to infinity. Such results are admittedly weak, but they are not even known for other nonconvex approaches. We would like to stress that the nearest-neighbor estimation of entropy *is not* needed to run the algorithm, it is only needed if one wants to plot the energy decrease.
>
> (3) A direct comparison to the approaches the reviewer mentions is difficult because we solve a different problem (those methods estimate a Waddington potential, and require $n_i>>1$). In contrast, gWOT solves the same estimation problem. Let us mention that optimal-transport-based approaches are compared to dozens of other approaches in the WOT paper [Schiebinger et al ‘19] and gWOT is compared to WOT in the gWOT paper.
>
> **Questions**
>
> (1) The (technical) proof of [1] would go through for Eq.(3) (the difference between the two objective functions is not important from a statistical viewpoint but our formulation is much more convenient for discretization/optimization). Rather than re-proving Thm. 2.1 with small changes, we are, in ongoing work, studying the sample complexity of (3) (i.e. a direct quantitative consistency result).
>
> (2) PDE (11) describes the evolution of the law of the marginals of the Mean-Field Langevin dynamics described by SDE (10), which we solve numerically by discretizing $\mu$ to a family of discrete particle clouds, as explained in Eq. (13) (the Laplacian term in (11) is a consequence of the noise term (13)), as is standard in the Mean-Field Langevin dynamics literature.
>
> (3) A computational complexity result to reach $\epsilon$-accuracy for the overall problem is beyond reach for the moment. With $1/\Delta t$ marginals discretized each into $m$ particles, we carry out Sinkhorn iterations for each pair of timepoints until an $\epsilon$ tolerance is reached in the dual Sinkhorn objective. We will expand the discussion in (Sec. 3.4) to mention that we have an *iteration* complexity of time $O(m^2/(\tau(\Delta t)^2\epsilon))$ using [Dvurechensky et al’. 18] complexity bounds for Sinkhorn.
>
> (4) Quantifying the dimension dependence is an important open question. Note that the scrna-seq dataset related to Fig. 4 has ambient dimension $d=20,000$ and we reduce to a space of $d=10$ dimensions by PCA when preprocessing the data.
>
> (5) As mentioned, this unbalanced extension is only introduced as a heuristic and is motivated by the practical problem of accounting for growth. We do not claim that there is theoretical support for this extension for the moment.

---

> > ### Comment · Reviewer_2ZNK · 2022-08-07
> > **Response to the authors**
> >
> > Thanks for the answers/clarifications. Based on them, I am willing to increase my score.

---

### Author Response · Authors · 2022-08-02
**revised version + fix to Thm. 3.3**

We thank the reviewers for their careful reading, comments and suggestions. We have replied to the specific comments of each reviewer separately. We have uploaded an updated version of the submission (and the supplementary material) with the important changes highlighted in red.

Shortly after submission, we have realized a mistake in the convergence theorem (Thm. 3.3) which is corrected in the new version. Our error was the following: the function $G$ from Eq.(7) is separately convex in each of its input $\mu_i$ but it is *not* jointly convex while the proof of Thm 3.3 requires joint convexity. Our fix is the following: instead of using Mean Field Langevin (MFL) to minimize the function $F_0= G+\tau H$, we apply it to $F_\epsilon= G + ( \tau + \epsilon)H$ and we interpret $F_0=G+\tau H$ as the convex term instead of $G$ ($F_0$ is indeed jointly convex, see Prop. D.1). This means that we need to add an additional entropic regularization term $\epsilon H$ in order to have exponential convergence. The original problem (with $\epsilon=0$) can be minimized using simulated annealing (see the new version of Thm. 3.3).
Note that in practice, we were already using simulated annealing (SA) to speed-up convergence (App. F), so the change does not significantly impact our numerical experiments. We will add experiments to compare between our previous version of SA (where $\tau$ decreases towards its final value and $\epsilon=0$) versus the one recommended by the new theory (where $\tau$ is fixed and $\epsilon$ decreases). There is a difference between these two procedures because $G$ also depends on $\tau$.

While this update slightly weakens our results, it also points at the interesting fact that MFL to minimize $F=G+\lambda H$ enjoys global convergence even when $G$ is non-convex, as long as $G+\tilde{\lambda}H$ is convex for some $\tilde{\lambda}<\lambda$ (at least in the compact setting).

---

### Meta-Review · Area_Chair_etuV · 2022-08-26

**Recommendation:** Accept
**Confidence:** Certain

**Metareview:**

This paper studies the challenging problem of inferring the trajectory of a stochastic process from sample observations of its marginals. Earlier work of Lavenant et al. introduced a consistent estimator based on an optimization problem over continuous time. The main contributions of this paper are in (1) introducing a discrete time variant, based on Schrodinger bridges and minimizing an entropy regularized optimal transport problem over the marginals rather than over path space. In Theorem 3.1 they show an interesting "Representer theorem" which shows that the two formulations are equivalent. And (2) they prove consistency of their estimator. In Theorem 3.3 they show exponential convergence. The main weakness is that the bounds are asymptotic in nature, and they do not get, for example quantitative bounds on how many particles are needed as the dimension grows. Overall this is still a nice contribution and seems like an accept.

**Award:**

No

---

### Decision · Program_Chairs · 2022-09-14

Accept